# Enhanced Adaptive Optics Control with Image to Image Translation

**Jeffrey Smith**[1]  **Jesse Cranney**[2]  **Charles Gretton**[1]  **Damien Gratadour**[2,3]

[1]School of Computing, Australian National University, Canberra, Australia
[2]Advanced Instrumentation Technology Centre, Research School of Astronomy and Astrophysics, Australian National University, Canberra, Australia
[3]LESIA, Observatoire de Paris, University PSL, CNRS, Sorbonne Universite, Université de Paris, 5 place Jules Janssen, 92195 Meudon, France

## Abstract

We aim to significantly enhance the science return of astronomical observatories, and in particular giant terrestrial optical telescopes. Observatories employ Adaptive Optics (AO) systems in order to acquire high sensitivity diffraction limited images of the sky. The incumbent "workhorse" for control of AO systems employs a linear real-time controller in a closed loop, with sensing of state performed via a (*Shack-Hartmann*) wavefront sensor (WFS). The actuators of a *deformable mirror* (DM) are driven, with the action performed in each iteration having a continuous representation as an array of DC voltages. The typical control regime is practical and scalable, nonetheless, there remains a residual uncompensated turbulence that leads to optical aberrations limiting the class of scientific assets that can be acquired. We have developed and trained a translational GAN model that accurately estimates residual perturbations from WFS images. Model inference occurs in 0.34 milliseconds using off-the-shelf GPU hardware, and is applicable for use in AO control where the control loop might be running at 500Hz. We develop an AO control regime with a second controller stage actuating a second DM controlled in an open loop according to the estimated residual turbulence. Using the open-source *COMPASS* tool for simulation, we are able to significantly improve the performance using our new regime.

## 1 INTRODUCTION

Adaptive Optics (AO) systems are an important component of astronomical imaging for large ground based telescopes, enabling the capture of high contrast images of faint objects in space. Aberrations due to Earth's atmospheric turbulence are a significant impediment to deep-space imaging, so the ability to estimate and compensate is critical. The scale of data requirements for this estimation problem increases quadratically with the telescope diameter, an ongoing problem as astronomers build larger telescopes to capture light from ever fainter objects, such as exoplanets and distant galaxies.

An AO system contains three main components: *(i)* a Deformable Mirror (DM) with a reflective surface that can be adjusted with an array of actuators to counteract some of the wavefront phase aberrations, *(ii)* a Wavefront Sensor (WFS) that collects information about the wavefront phase, and *(iii)* a controller that interprets the wavefront sensor observations and computes a control action to drive actuators to update the DM state. Wavefront phase estimation is required because atmospheric turbulence is a stochastic process typically evolving over a few milliseconds time frame. The real-time control of an instrument's mirror incorporates estimation at high-frequency – e.g. 500Hz. The WFS is used to capture the instantaneous state of the phase into intensity variations in an image. In the case of the Shack-Hartmann concept, the telescope aperture is split into sub-regions, called sub-apertures, and an image of the reference guide source is created for each of these sub-apertures and captured by a camera. A centroider algorithm is then used to estimate the spot displacements, in each of these sub-apertures, with respect to a reference position. These displacements are directly related to the local slopes of the wavefront [Roddier, 1999].

Existing real-time closed-loop mirror control uses displacement information. A key drawback is that all such control regimes neglect non-linear, high-order wavefront information captured on the WFS. This is an important limitation of existing instruments. It is highly desirable that non-linear information is available in real-time. This enables the conception of novel control regimes, to increase performance of the AO system and thereby the image quality at the focus of the telescope.

*Accepted for the 38th Conference on Uncertainty in Artificial Intelligence* (UAI 2022).

## 1.1 CONTRIBUTIONS

Building on a translational GAN network architecture motivated in a range of computer vision applications, we have developed and evaluated: *(i)* A new method for real-time phase estimation from wavefront sensing data in AO, and *(ii)* a new robust control regime that is able to significantly enhance instrument performance by leveraging our innovative approach to phase estimation. Compared to the model-based state-of-the art, our network-based approach to phase estimation is: *(i)* fast enough for use in real-time control, *(ii)* avoids strong (unrealistic) assumptions about the nature of the stochastic process driving atmospheric turbulence and system geometry, and *(iii)* conceptually simple and based on established machine learning technology. Specifically, we use a translational CNN to infer the phase directly from the Shack-Hartmann WFS image. Our method takes advantage of high frequency information available in the Shack-Hartmann WFS that is not accessible with existing estimation methods. On the control side, we develop a *GAN Assisted Open Loop* (GAOL) AO design. To our knowledge this is the first AO concept to apply accurate wavefront estimates in a way that is amenable to real-time control of AO systems with real-world operating parameters. We expose higher-order information in the control loop via our phase estimates, enabling control of DMs with relatively high actuator counts, and thereby enable AO systems that effectively compensate for high-frequency turbulence. The simulation experiments demonstrate that our approach robustly leads to substantially improved performance—specifically a 70 nanometer RMS reduction in *wavefront error*, which translates to an improvement which materially adds to the range of science tasks that an instrument can perform—in a range of atmospheric conditions when compared to incumbent AO methods.

The paper is structured as follows. First, we discuss the AO setting and current methods. Secondly, we describe COMPASS – a state-of-the-art GPU accelerated AO simulation package. Thirdly, we describe our approach to using image-to-image CNNs for phase recovery in our setting. Fourthly, we describe current methods in wavefront phase estimation and control methods for AO systems. Finally we present a detailed analysis of experimental results comparing our approach to a classical AO systems control approach.

## 2 ADAPTIVE OPTICS BACKGROUND

The goal of AO is to obtain a sharp image of an observed target. Any perturbation of the incoming wavefront creates aberrations in the image, which reduces the contrast of the observation – this translates as blur. A perfectly unaberrated image of a point source obtained through a circular aperture telescope will produce a *point spread function* (PSF) that is an *Airy disk* – i.e., image quality and resolution are only limited by the diffraction of the telescope aperture. Aberrations (e.g., from turbulence) perturb the wavefront by introducing optical path differences between the different points of the telescope aperture, affecting the PSF, and therefore image quality.

The PSF is related to the wavefront phase as the absolute square of the Fourier transform of the complex electromagnetic field. This is an important relationship and is not directly invertible from PSF to phase, shown in Eqn. 1.

$$\text{PSF} = |\mathcal{F}(\text{amplitude} \cdot e^{\text{i} \cdot \text{phase}})|^2 \qquad (1)$$

In an AO system a DM is controlled in real-time to compensate for aberrations. The compensation made using incumbent control regimes is imperfect, leading to residual error. The residual error (AO loop error) is made up of several contributing sources and is dealt with in detail through an error budget estimation. The error budget describes the total AO loop error in terms of components for Bandwidth error, Anisoplanatism, Aliasing, Noise, Wavefront measurement error, mode filtering and fitting error. All of these error sources contribute to a decrease in image quality at the output of the telescope, and are described and incorporated into the state-of-the-art residual wavefront error estimation by Ferreira et al. [2018a].

The Shack-Hartmann WFS used in AO systems is designed to take the wavefront phase information and encode it as an intensity image distributed over small sub-regions of the aperture. It does this with an array of small lenslets that focus the aperture sub-region onto a sensor, creating a spot that is tilted off axis by the average slope of the incoming wavefront sub-region in two dimensions. Figure 1 shows the one dimensional case and how the aberrated wavefront of a sub-region moves the focal point on the sensor off center. This displacement gives an indication of the average slope of the area of the wavefront covered by the sub-aperture. From the sensor image for all sub-apertures, centroider algorithms are used to find the centre of the spots and so a granular map of slopes is created and passed to the DM control system.

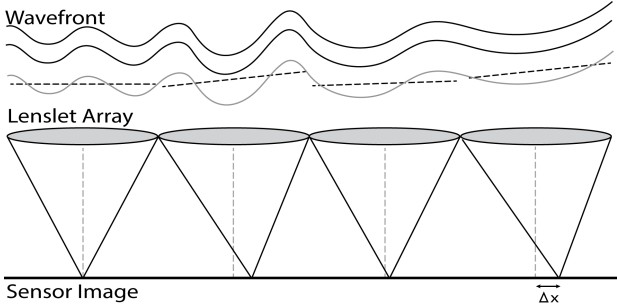

Figure 1: Shack-Hartmann WFS lenslet diagram showing displacement of spots due to wave from perturbations

Slope measurements made from centroider data are inherently lossy due to limits on sub-aperture size. Non-linear

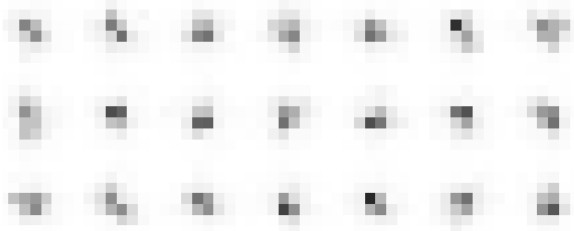

Figure 2: Shack-Hartmann WFS lenslet spots

information is lost when the algorithm picks the centroid of the spot for each sub-aperture, reducing the image to points on an x, y plane. Figure 1 shows the off axis measurements ($\Delta x$) that are used to measure the wavefront sub-aperture slopes, where the higher-order wavefront information is lost. The size of the lenslets limits the spatial frequency that can be measured, behaving like a low pass filter.

The sensor image captured at each subaperture corresponds to a low fidelity PSF, where the captured irregular patterns of light intensity correspond to a representation of higher-order information about the wavefront. A depiction of such patterns is given in Figure 2, which shows a portion of a simulated Shack-Hartmann WFS image. For the intuition about what is being lost using centroider algorithms, Figure 1 gives a simplified 1D schematic of the wavefront, lenslet array and sensor image. The dashed lines drawn in the turbulent wavefront above lenslets represent the gradient measured by a centroider, here clearly missing out important details about high frequency turbulence. It remains an open question in astronomy, to quantify exactly how much information is being lost in this setting depending on the actual instrument design (e.g. number of sub-apertures, number of pixels per sub-apertures, measurement wavelength, etc.). Model-based approaches to phase retrieval rely on detailed physical models to recover phase information from centroider slopes only – i.e. the irregular patterns from the WFS are not used. Our approach is model free, using precisely that information to recover phase.

**AI for Adaptive Optics:** AO systems have been used to compensate for atmospheric turbulence since the late 1980s, when the available computer technology was first able to match the requirements for controlling the available DM technology. Since then, efforts to improve wavefront estimates have been ongoing both in model-based statistical estimation and other techniques using advances in Artificial Intelligence.

Some AI based techniques for improving AO systems have been investigated with many studies using slope estimates from centroider data [Swanson et al., 2021] and typically making wavefront estimates by predicting the weights of a small number of low order, linearly independent Zernike modes [Guo et al., 2006; Weddell and Webb, 2007] that

can be added to create the wavefront phase. Using the slope estimates from centroider algorithms limits WFS data to low order information, as these wavefront slope estimates filter out higher order information captured on the Shack-Hartmann WFS. Sensor-less methods with Convolutional Neural Networks (CNNs) [Guo et al., 2019] make estimates directly from the PSF in the AO loop, rather than from a WFS, can avoid the loss of some higher-order information but are best suited to low turbulence conditions and small telescope settings.

**COMPASS Simulation Software:** The COMPASS AO simulation software [Ferreira et al., 2018b] simulates atmospheric conditions, telescope and AO system to create accurate simulated residual wavefront and WFS images used to train our CNNs. COMPASS is a GPU accelerated AO loop simulator with a comprehensive API that allows simple integration with Python code. Highly detailed parameter information can be input to generate specific atmospheric conditions and other AO loop characteristics such as sensor noise and control loop delay. This is perfect for generating training data for CNNs and also for testing ranges of conditions for inference performance with trained models. See Figure 3 for sample of simulated data.

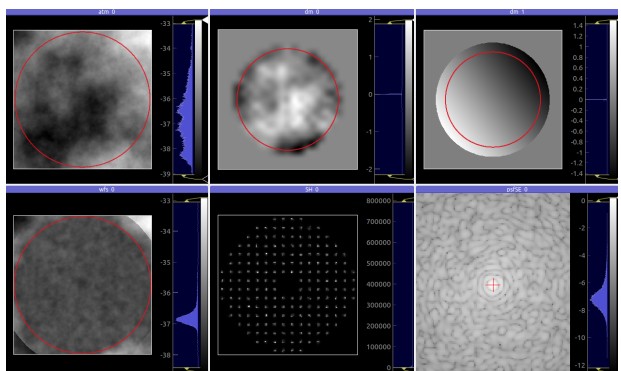

Figure 3: COMPASS Artifacts - typical data available through simulator. The top left image shows the simulated atmosphere, top-middle and top-right shows the deformable and tip tilt mirrors respectively. The bottom left image is the residual wavefront phase and the bottom middle is the Shack-Hartmann WFS image – these are the two images we will use for training data. The bottom-right image is the short exposure PSF image in log scale.

## 3 AO CONTROL

Typical AO systems consist of three main components, i.e. the DM, WFS and controller. Such systems have many possible configurations that are designed for purpose. The incumbent design that we are using for a benchmark is a Single Conjugate Adaptive Optics (SCAO) system with a Shack-Hartmann WFS. The command signal, $u_k$, applied to the

DM(s) at time $t = kT$ (where $T$ is the AO control period) is derived from the WFS slope vector $s_k$, and the previous DM commands $u_{k-1}, u_{k-2}, \cdots, u_{k-d}$, where $d$ is the AO control loop latency in frames. For the remainder of this paper, we assume a 2-frame delay (without loss of generality). This delay is the source of *bandwidth error* in an AO system, and is the combination of several effects: integration time on the WFS camera, frame transfer and read-out on the detector, pixel data transfer to the Real-Time Control (RTC), RTC pure delay, DM rise time and DM zero-order hold.

For its convenient stability properties and robust performance, we employ a *Pseudo-Open-Loop Control* (POLC) scheme [Ellerbroek and Vogel, 2003; Piatrou and Gilles, 2005], which has a recursive control law in the form:

$$u_0 = \mathbf{0},$$
$$u_k = (1 - g)u_{k-1} + gRDu_{k-2} + gRs_k, \qquad (2)$$

where $1 - g$ and $g$ are the input and output coefficients (respectively) of a first-order IIR filter, and can be tuned to obtain a desired temporal cut-off frequency [Cranney et al., 2020]. The matrix $D$ is the interaction matrix between the WFS slopes and the DM commands. The matrix $R$ is the linear reconstructor matrix, which takes WFS slopes, and computes the (filtered) commands that would optimally regulate those measurements.

**Our GAOL Control Regime:**   In the following section, we develop a nonlinear network-based method to extract high resolution wavefront estimates from closed-loop WFS pixel data (beyond slope-only estimation). This is the basis of our GAOL control regime, in which such wavefront estimates are projected onto a DM in order to provide an improved wavefront correction at a higher-order than possible using a linear slope-based controller. The most direct method to evaluate the performance of such a scheme in AO is to add a second DM (with a high actuator density) before the science camera, but after the first DM and WFS, and to control this second DM in open-loop using the output of the nonlinear estimator. The use of an open-loop DM has several advantages and disadvantages that will not all be discussed in detail here. The most notable are that the second DM has guaranteed *bounded-input bounded-output* stability, assuming that actuator creep is negligible, coming at the financial cost and opto-mechanical burden of having multiple DMs.

In principle the control of all DMs could be done on an orthogonalised set of modes, with the modes able to be linearly controlled put on the first DM, and the remainder allocated to the second DM (i.e., a *Woofer / Tweeter* pair), [Gavel and Norton, 2014]. For our GAOL approach, we instead match the first DM to the WFS geometry (so-called *Fried geometry*), and increase the resolution of the second DM beyond this limit.

The closed-loop linear control law in this regime remains

identical to Eqn. (2), and the open-loop nonlinear control law operates in parallel with the following structure:

$$u_0^{\mathrm{nl}} = \mathbf{0},$$
$$u_k^{\mathrm{nl}} = (1 - g^{\mathrm{nl}})u_{k-1}^{\mathrm{nl}} + g^{\mathrm{nl}}R^{\mathrm{nl}}\hat{y}_k, \qquad (3)$$

where $u_k^{\mathrm{nl}}$ is the nonlinear command signal to be applied to the second DM, $\hat{y}_k$ is the high resolution network-derived estimate of the residual phase error after the first mirror has acted, $R^{\mathrm{nl}}$ is the high resolution reconstructor which takes the estimated phase and projects to the DM space, and $1 - g^{\mathrm{nl}}$ and $g^{\mathrm{nl}}$ are the input and output coefficients (respectively) of a first-order IIR filter (*cf.* Eqn. (2)).

# 4   GAN FOR PHASE ESTIMATION

We estimate the residual phase by adapting an artificial neural network for Image to Image translation [Isola et al., 2017], where Figure 4 gives a visual breakdown of the network. This design is a conditional Generative Adversarial Network (cGAN), with the translational encoding performed by a UNet and the adversarial training performed using a Markov discriminator. The network learns to take an input of a Shack-Hartmann WFS image and output the inferred wavefront phase.

To motivate the UNet generator component of the network, it can be compared to the similar and widely used auto-encoder – a CNN – that is used for image transformation. An auto-encoder encodes an image to some latent variable through successive convolutional layers, and then through deconvolutional steps generates a new image from the latent variable. An auto-encoder learns to transform images minimising a reconstruction loss and can be used for several applications. For our purposes an auto-encoder is not ideal, because it is deterministic by design, and because we need to preserve some structure from the original image in our application, such as spatial relationships for translation. The UNet design adds skip connections, where information from layers of the encoder is transported to corresponding decoder layers via concatenation, allowing for some structure from the input image to be preserved. Because we aim to translate WFS images from sensor data with incomplete information we cannot map from image to image in a deterministic manner as there will be many possible wavefront phase images that could be represented by each input image. To avoid the deterministic nature of the auto-encoder (and UNet) structure, some stochastic process needs to be added to allow for variability in the output. This is accomplished by introducing noise to the network via network dropout ($z$), where Gaussian noise is ineffective because this approach learns to filter it.

Considering the Variational Auto-Encoder (VAE) as an alternative - it also has the ability to create varied translations from an input image and does not have deterministic out-

comes due to encoding and sampling from distributions. The output images from a VAE tend to be blurry and faint, which is not ideal for our application, as we find this occurs in important regions of the wavefront phase. The loss function for a VAE must be carefully designed, which is additionally very difficult to do in practice. By contrast, a GAN [Goodfellow et al., 2014] has the benefit of learning a loss function, and so simplifies the loss design problem associated with VAEs, as well as tending toward sharper output images, while adding complexity to the network with the addition of a CNN classifier forming the discriminator network that is trained simultaneously with the generator.

A *conditional* GAN improves the performance of the GAN by including a 'semantic-image' in the discriminator as a paired image with either the "real" or "fake" image, which acts as a label for the distribution generated, adding supervision which further improves the sharpness and accuracy of the translated image. The discriminator in Figure 4 is a PatchGAN discriminator, also known as a Markov discriminator [Li and Wand, 2016]. This discriminator architecture operates by classifying local image regions, and is broadly motivated in computer vision applications due to the speed of inference (i.e. local inference is relatively fast), and the quality of PatchGAN architectures in preserving complex image detail, such as texture. The discriminator component is a convolutional classifier, trained simultaneously with the generator, with the objective to maximise the value of Eqn. 4.

The overall objective function (Eqn. 5) combines the cGAN loss (Eqn. 4), with the $L1$ reconstruction loss terms (e.g., of the form in Eqn. 6) that provide strong guidance to learning of low frequency structure. Note that a second reconstruction loss term is added in Eqn. 5 to reinforce the reconstruction loss where the network underestimates the upper and lower extreme phase values $G_M$ (or G 'masked'). These regions of the wavefront with the largest perturbations substantially impact the important metrics in our setting, and our ablation testing of this parameter shows this term is required.

The approach is to simultaneously train: *(i)* a generator, $G(x, z)$, that models the distribution of wavefront phases consistent with the input WFS image $x$ – i.e. $z$ is a noise term, specifically dropout noise, and *(ii)* a discriminator, $D(x, y)$, that estimates the probability that a pair $(x, y)$, comprising a wavefront phase image $y$ and a corresponding WFS image $x$, are "real".

$$\mathcal{L}_{cGAN}(G, D) = E_{x,y}[\log(D(x, y))] \\ + E_{z,x}[\log(1 - D(x, G(x, z)))] \quad (4)$$

$$G^* = \arg\min_G \max_D \mathcal{L}_{cGAN}(G, D) \\ + \lambda \mathcal{L}_{L1}(G) + \lambda_M \mathcal{L}_{L1}(G_M) \quad (5)$$

$$\mathcal{L}_{L1}(G) = E_{x,y,z}[|||y - G(x, z)||_1] \quad (6)$$

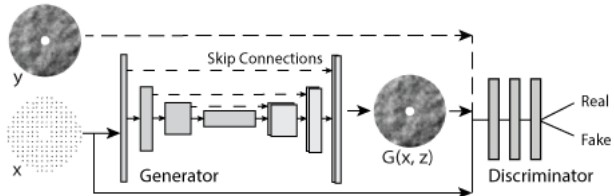

Figure 4: cGAN architecture – UNet / PatchGAN

**Network architecture:** We adapted our network architecture and code from Isola et al. [2017], where much of the architecture details remains the same and we will follow the same labeling conventions. $Ck$ denotes a Convolution-Batchnorm-ReLU layer with $k$ filters and $CDk$ denotes a Convolution-BatchNorm-Dropout-ReLU layer. Dropout rates, stride, downsample scaling, and upsample scaling, are all determined as per the literature mentioned above. Refer to the literature for any other parameters.

**Generator architecture –** Our network uses a UNet generator, consisting of an encoder, a decoder and skip connections between some layers as shown in the following layer structures:

*UNet encoder:*

$C64$-$C128$-$C256$-$C512$-$C512$-$C512$-$C512$-$C512$

*UNet decoder:*

$CD1024$-$CD1024$-$CD1024$-$C1024$-$C1024$-$C512$-$C256$-$C128$

**Discriminator architecture –** The Markovian Discriminator architecture by layer:

$C64$-$C128$-$C256$-$C256$

**Data Transformations:** The raw data from astronomical instruments and simulators requires some transformation to be amenable to the translational architecture we have just discussed. First, the piston mode—i.e., a constant phase shift across the full aperture—is removed from the residual phase data, since it is not measured by the WFS. This is done by subtracting the average value of the wavefront phase that is inside the pupil from the phase array. Second, the residual phase data is normalised to sit in the range $[0, 1]$ by dividing through by a constant value so that the amplitude can be restored by multiplying any inferred image from the network by the same constant. The normalisation factor is a tuning parameter, with high values scaling small wavefront errors too much, leading to mode collapse. When normalised perfectly, so the largest value in the training data is exactly 1, we find the trained network does not perform well in generalisation evaluations – For example, where we infer a residual phase in turbulence unseen during training. In our work we have set this factor to 10, leaving some headroom over the minimum required value, apprx. 7. The WFS image is also normalised, again dividing by a constant

value which is slightly above the maximum. The WFS scale is preserved because the WFS amplitude information, along with the shape of the WFS spots, is the additional nonlinear information captured using our estimation method. In all cases our networks use a constant scaling factor of 1.2 million for WFS images. This value is chosen according to the magnitude of the guide star, with our simulations using a fixed guide of magnitude 10.

# 5 EXPERIMENTAL RESULTS

**COMPASS Parameters:** Parameters for simulation were selected to demonstrate performance for realistic large telescope AO loop scenarios. The degree of turbulence is defined by the so-called Fried parameter, $r_0$, which is a measure of the coherence scale of the turbulence [Roddier, 1999] and depends on the wavelengths we are observing. Typically the real-world operating conditions for $r_0$ are in the range of $0.16m$ at visible light wavelengths for the lower range of atmospheric turbulence to $0.05m$ for extreme conditions. For the purposes of this study we have selected a typical $r_0$ value of $0.10m$ for GAOL design analysis and a range from $0.06m$ to $0.16m$ for robustness testing to atmospheric turbulence.

AO loop data has been simulated for a typical wind speed of 10 $ms^{-1}$, and 50000 sample image pairs each from data generated with $r_0$ values in $[0.093m, 0.15m, 0.20m, 0.25m, 0.30m, 0.35m, 0.40m]$. We thereby provide the network with a range of turbulence scenarios to learn from, to aid robustness for estimation in variable atmospheric turbulence that would be expected for on-sky operating conditions. The upper limit of 0.093 was selected as it corresponded with a pupil size of 512 pixels. The training and evaluation atmospheres are different. In particular, when interrogating network model performance, in a control setting below and otherwise, we use simulations that are seeded uniquely, and therefore are of atmospheres not seen during training. See Table 1 for simulation parameters.

**GAN Network Parameters:** We adapted the network architecture and code from Isola et al. [2017] to use our revised loss regime. Both Generator and Discriminator networks used 64 filters. The Generator performs better with 64 filters over trials run with 32 or 16, however this comes with a computational cost as the number of parameters is significantly increased which in turn increases training time and hardware memory requirements. Our training dataset is generated using COMPASS, and consists of $350,000$ image pairs. In training these are selected at random and we use a batch size of 1.

Loss parameters were manually optimised for our setting, and indeed we require a second loss term. The loss regimes from the literature lead to models that underestimate ex-

Table 1: Simulation parameters

| Telescope Parameters | |
|---|---|
| Diameter | $8\ m$ |
| Simulated Atmospheric Parameters | |
| Number of Layers | 1 |
| $r_0$ | $0.10\ m$ |
| Wind Velocity | $10\ ms^{-1}$ |
| Target Parameters | |
| Wavelength $\lambda_t$ | $1.65\ \mu m$ |
| WFS Parameters | |
| Number of sub-apertures | 16 x 16 x 8pix |
| Wavelength $\lambda_{wfs}$ | $0.5\ \mu m$ |
| AO Parameters | |
| Loop frequency | $500\ Hz$ |
| Delay | 2 frames |
| Integrator Gain | 0.4 |
| # Frames per experiment | 2000 |
| DM Parameters | |
| Number of DM actuators (Woofer) 1 tip-tilt mirror | 17 x 17 |

treme maximum and minimum phase, and this has substantial repercussions in our application.

Our method has a second L1 loss term, which increases the weight of L1 loss for regions of phase screens with boundary values. This extends Eqn. 5 with the additional masked L1 loss term where $G_M$ defines the region of extremes masked in the generated image $G$, and $\lambda_M$ is the weighting coefficient for the added masked loss term. All experimental results in this paper use a single trained network with hyperparameters $\lambda = 150$ and $\lambda_M = 30$.

## 5.1 AO LOOP SIMULATIONS

One of the benefits of the GAOL design we propose is the simplicity in extending a typical SCAO linear closed controller by adding a second DM before the science camera. This mirror is controlled by the estimates from the GAN and operates in open loop – I.e., the effect of the second DM is not fed back to the closed loop. For ease of reference we will refer to the first DM as the 'Woofer' DM, in reference to the low order information it corrects for, and the 'Tweeter' DM is that driven by the GAN estimates that allow for high-order corrections [Gavel and Norton, 2014].

Referring to Figure 5, the simulated GAOL AO system consists of a linear controller in closed loop operation, to which we add a DM Tweeter and controller before the science camera (highlighted). Our translational network makes estimates that include high-order wavefront information that is able to drive the Tweeter DM with a higher number of actuators. The input to estimation is the same WFS image

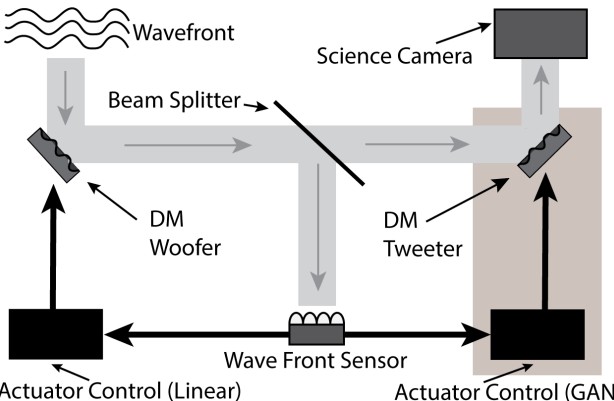

Figure 5: GAOL AO Loop diagram

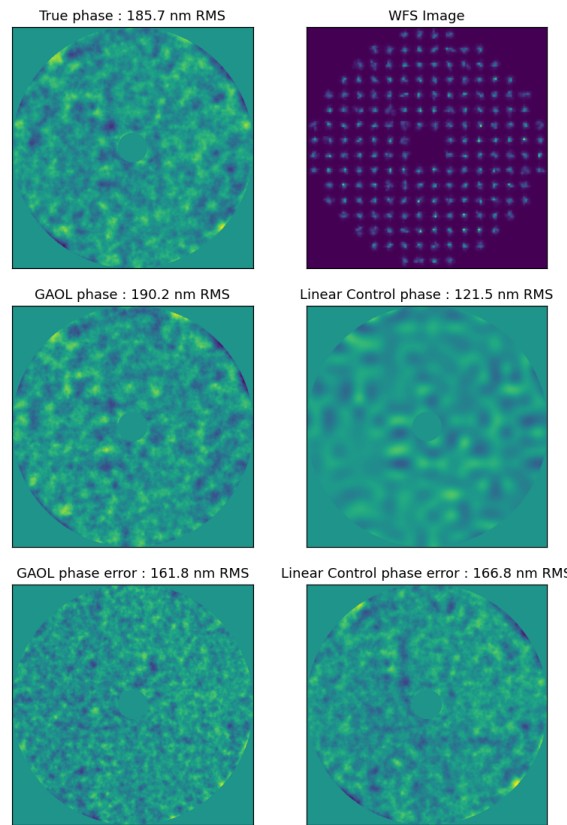

Figure 6: Phase images from GAOL AO design (Linear and GAN Residual Phase Estimate steps) compared with the simulated true residual phase image in the AO Loop, all images after 2000 iterations. Min to max range values for phase images: True 2036nm, GAOL 1971nm, Linear 961nm, GAOL Error 2926nm, Linear Error 2422nm

that is used by the closed loop linear control section of the system.

For our experiments we contrast the performance of the GAOL design against the Linear controller, and we vary the number of actuators in the open-loop (Tweeter) DM. The motivation of such an experiment is that if the GAOL controller is able to outperform the linear one, then the GAN must be able to more accurately infer information about the wavefront phase than a slope-based linear reconstruction scheme can. As AO loops take time to stabilize, we run each experiment for 2000 iterations (4 seconds) before recording the *long exposure* (LE) SR from COMPASS. The average RMS wavefront error can be estimated from the LE SR, given simulation parameters shown in Table 1.

## 5.2 EVALUATION METRICS

For the purposes of evaluating the performance of GAOL control we choose a benchmark of the incumbent SCAO control regime that is easily simulated with COMPASS. Contrasting the performance of our novel control variants in terms of Strehl Ratio (SR) and RMS wavefront error (WFE) will demonstrate relative performance in domain specific metrics. The realisation of turbulence for each simulation is identical, so performance can be compared directly.

The SR measures the light intensity at the core of the PSF relative to a diffraction limited PSF, so a SR of 100% implies an aberration-free image. The RMS WFE is independent of imaging wavelength, and determines the quantity of residual wavefront phase of a given wavefront. A RMS WFE of 0 nanometres (nm) implies an aberration-free wavefront, which produces an aberration-free image. Comparisons of RMS WFE are calculated as discussed in Ross [2009].

## 5.3 RESULTS AND ANALYSIS

The GAOL method is a two step process, leveraging the incumbent closed loop linear controller that relies on slope

measurements through centroider techniques on the Shack-Hartmann WFS image. The phase estimate from the slope measurements is then used to find a control solution to drive the closed loop Woofer DM state. The GAOL method extends the linear controller by passing the closed loop WFS image through the GAN to infer the residual phase, and then use this inferred phase to drive a second (Tweeter) DM to further reduce the WFE. In this way, we can keep a closed loop residual phase that is unaffected by the GAOL control for future iterations of the AO loop.

For comparison of the GAOL control method with the linear control baseline, Figure 6, shows for a single AO loop iteration, the true residual phase (top-left) seen by the WFS (top right), after interacting with the Woofer DM. Residual phase estimates are made from the WFS observation – one with the linear controler (mid-right) and the other inferred using the GAN (mid-left) to generate a residual phase estimate for the Tweeter DM. Each of these phase estimates is shown with the respective WFE shown in the image below them. For the baseline linear control case, we can see from the

Table 2: Performance (in Strehl ratio at 1650 nm) of each control scheme. Each controller is evaluated under each system configuration, with a fixed $r_0$ of 0.10m ($\Delta$WFE indicates the relative RMS wavefront error in nanometres for each method compared to the linear controller)

| AO System | Linear Control LE Strehl | GAOL Control LE Strehl | GAOL Control $\Delta$WFE | Oracle Control LE Strehl | Oracle Control $\Delta$WFE |
|---|---|---|---|---|---|
| No Open Loop DM | 62.20% | N/A | N/A | 73.76% | 108.42 |
| Open-Loop +0 (17x17) | 62.37% | 67.33% | 72.64 | 73.76% | 107.55 |
| Open-Loop +1 (18x18) | 63.96% | 69.87% | 78.07 | 75.76% | 108.06 |
| Open-Loop +3 (20x20) | 64.06% | 70.67% | 82.82 | 79.17% | 120.85 |
| Open-Loop +7 (24x24) | 66.77% | 72.45% | 75.03 | 83.99% | 125.79 |
| Open-Loop +15 (32x32) | 69.39% | 73.27% | 61.25 | 89.21% | 131.63 |
| Open-Loop +31 (48x48) | 68.55% | 72.13% | 59.25 | 93.41% | 146.08 |

Table 3: Robustness to atmospheric turbulent conditions ($r_0$) of optimal setting, compared to linear control baseline (both with +7 actuators on the tweeter DM, in Strehl ratio at 1650 nm. $\Delta$WFE indicates the relative RMS wavefront error in nanometres compared to the linear controller)

| Fried parameter $r_0$ (m) | Linear Control +7 actu (24x24) LE Strehl | GAOL +7 actu (24x24) LE Strehl | GAOL +7 actu (24x24) $\Delta$WFE |
|---|---|---|---|
| 0.06 | 38.72% | 44.00% | 93.89 |
| 0.08 | 56.43% | 62.04% | 80.85 |
| 0.10 | 66.77% | 72.45% | 75.03 |
| 0.12 | 73.34% | 79.24% | 73.05 |
| 0.14 | 77.77% | 83.73% | 71.36 |
| 0.16 | 80.94% | 86.89% | 69.94 |

phase estimate that some limited estimation of the residual is possible, and this translates to a modest improvement in the WFE. The residual phase estimate inferred by the GAN however is clearly able to interpret higher-order wavefront information from the WFS, and is a close fit to the true residual. Comparing the two residual phase errors, on the bottom row of Figure 6, it is clear from the much finer grain phase perturbations in the GAOL phase error that the linear controller misses the high-order information inferred by the GAN, and that the GAN estimate can be used to greatly reduce the residual phase error of the closed loop linear controller when applied to the Tweeter DM in open loop for real-time control.

Table 2 shows the results of the comparison. Each column of the table corresponds to a different method used to drive the DMs in the AO system. Each row of the table corresponds to a different AO system configuration, where the key difference between each configuration is the number of actuators in the 'Tweeter' DM. Each DM has all of its actuators uniformly spread across the pupil, so a larger number of actuators allows a higher-order correction of the wavefront by this DM.

Consider the 'Oracle Control', where all information about the wavefront is known to a perfect accuracy. The first and second rows correspond to systems with DMs of equivalent order, but with one DM in closed-loop with the SHWFS and one DM in open-loop. The Oracle performs equivalently on each of these systems. For the remaining rows, the Oracle monotonically improves, due to the higher degree of freedom in its actuator space.

The 'Linear Control' is also allowed to actuate with a higher degree of freedom for each row in the table, though it can only use measurements derived from the SHWFS, which has a limited sensing resolution based on its geometry. Observing that the WFE of the Oracle with respect to the linear control monotonically increases, it is clear that the linear controller is not able to fully utilise the additional actuators of the open-loop DM.

The 'GAOL Control' is designed such that the open-loop DM is only controlled by the GAN-inferred wavefront. So the observation that GAOL consistently outperforms Linear control indicates that the GAN is indeed able to more accurately infer wavefront information than is possible with a slope-based linear scheme. This observation suggests two important results:

1. The GAN is able to more accurately infer wavefronts than a slope-based scheme when DM is geometrically matched to the WFS (since the second row of the table shows an improvement for the GAN, despite not having any extra actuators across the pupil),

2. The GAN is able to infer higher-order spatial information relative to the slope-based scheme (since the $\Delta$WFE improves, at least up to the +3 actuator sys-

tem).

We see that for very large actuator counts the Linear and GAOL control schemes begin to degrade in performance. This is likely due to numerical issues involved in controlling many more actuators than available measurements, though it is worth noting that at the point where the performance begins to degrade, the open-loop DM has 9 times as many actuators as the number of subapertures in the WFS.

From Table 3 it is clear that the GAOL control loop consistently delivers large performance gains over the linear control method for all $r_0$ values, where the performance improves slightly as the turbulence increases. This demonstrates robustness to changes in the degree of atmospheric turbulence, where a single pre-trained GAN can infer for a wide range of conditions, removing the need to match networks to conditions.

## 6 DISCUSSION & FUTURE WORK

We have shown that with CNN Image to Image Translation we can utilise high-order information from the Shack-Hartmann WFS in an AO loop that is inaccessible with centroider algorithms applied by current practical wavefront estimation strategies. With this high-order data, our translational network can accurately estimate the wavefront from just the WFS image. By controlling a second DM in open-loop using the GAN estimated wavefront, we can further reduce the residual phase by at least 70 nm RMS compared to the best available linear controller. Performance of this GAN-based method is shown to be robust over a wide range of real-world turbulence conditions. [1]

With access to higher-order Shack-Hartmann WFS data, we can control more DM actuators with less Shack-Hartmann WFS sub-apertures. This translates to a substantial impact on how future AO systems will be designed – either as a cost saving (using cheaper components), or a science enabler (increasing sky coverage, and building a more potent systems providing extreme AO correction—high actuators count—with same sky coverage as current instruments).

We are the first to design a highly accurate method of residual wavefront estimation and apply it to real-time control. Specifically, the time-to-solution is in the right regime, with an average inference time of 0.34ms on a desktop GPU. With optimization and dedicated hardware [Perret et al., 2016; Gratadour et al., 2020], and the COSMIC framework platform [Ferreira et al., 2020], there is potential for hard real-time AO control using wavefronts inferred from pre-trained networks. Future lab experiments will allow for verification of loop control with real sensor equipment.

While these experimental results are just the beginning for this project, they are a proof of concept that image to image

---

[1]Code at `https://github.com/GANs4AO/`.

---

translation with CNNs can efficiently and accurately estimate the residual wavefront in AO systems and be easily applied in simulation with few additional components. The speed and simplicity, combined with demonstrated performance benefit of our method is of great practical interest to the AO community and shows a lot of promise for the implementation requirements of the in-construction ELT, applying eXtreme Adaptive Optics (XAO) to search for exoplanets[Kasper et al., 2021]. Additionally, our method opens the door to upgrading existing AO systems to provide better turbulence compensation while preserving sky coverage, since the number of sub-apertures can remain unchanged. In particular, currently contemplated upgrade projects like SPHERE+ as discussed by Beuzit et al. [2018] would clearly benefit from this approach by providing a cost effective way to increase the AO performance and enhance the science return of this existing instrument.

**Author Contributions**

Damien Gratadour conceived the idea of applying AI to the AO estimation problem for PSF reconstruction and real-time AO control, and supervised experimental methods and analysis of results as applied to the AO setting.

Charles Gretton conceptualised the application of advanced CNN methods and supervised experimental methods and analysis of results in the AI setting.

Jesse Cranney designed and implemented the simulated AO loop configurations, including COMPASS configurations and provided deep, applied knowledge of AO methods, COMPASS scripting and analysis.

Jeffrey Smith applied the Image to Image Translation concept for the project context and conducted research and experimentation for estimation techniques, and applied these to simulated AO control experiments.

The paper was written by Jeffrey Smith, with guidance and input from all co-authors excluding the AO Control section written by Jesse Cranney.

**Acknowledgements**

Many thanks to Florian Ferreira for donating his time and knowledge assisting with COMPASS, Felipe Trevizan for guidance on manuscript editing and Mark Burgess for reviews and feedback.

This work was supported in part by Oracle Cloud credits and related resources provided by the Oracle for Research program.

This research was undertaken with the assistance of resources from the National Computational Infrastructure (NCI Australia), an NCRIS enabled capability supported by the Australian Government.

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
