# OpenReview forum: "Enhanced Adaptive Optics Control with Image to Image Translation"
_auai.org/UAI/2022/Conference — UAI 2022 Poster_

### Official Review · Reviewer_dcdU · 2022-03-23

**Q2(1) Originality/Novelty:** 1
**Q2(2) Significance/Impact:** 1
**Q2(3) Correctness/Technical Quality:** 2
**Q2(6) Clarity Of Writing:** 3
**Q6 Overall Score:** 4
**Q8 Confidence In Your Score:** 5

**Q1 Summary And Contributions:**

The author developed a translational GAN model that accurately estimates residual perturbations from WFS images.

**Q2 Assessment Of The Paper:**

More detailed information regarding each of these aspects is given below:

**Q2(4) Quality Of Experiments (Optional):**

2: Fair: The experimental evaluation is weak: important baselines are missing, or the results do not adequately support the main claims.

**Q2(5) Reproducibility:**

2: Fair: Key resources (e.g., proofs, code, data) are unavailable but key details (e.g., proof sketches, experimental setup) are sufficiently well-described for an expert to confidently reproduce the main results.

**Q3 Main Strengths:**

Using deep learning in the field of optics is a good way

**Q4 Main Weakness:**

1.	The experiment is incomplete and there is no extensive comparison with the previous algorithm.
2.	The experiment is incomplete and there is no extensive comparison with the previous algorithm.
3.	The arrangement in your article seems unreasonable especially in the last section.


**Q5 Detailed Comments To The Authors:**

1.	you can use the new network architecture to rebuild the part where you use deep learning.
2.	he writing skill needs to be strengthened.
3.	More experiments should be done to prove your conclusion.


**Q7 Justification For Your Score:**

This conference focuses on the knowledge in the field of depth learning ,but the author’s work in the field of deep learning is out-of-date. SO I value your weaknesses more than your strengths.

**Q9 Complying With Reviewing Instructions:**

1: Yes.

---

### Official Review · Reviewer_rMiF · 2022-04-10

**Q2(1) Originality/Novelty:** 3
**Q2(2) Significance/Impact:** 3
**Q2(3) Correctness/Technical Quality:** 2
**Q2(6) Clarity Of Writing:** 2
**Q6 Overall Score:** 6
**Q8 Confidence In Your Score:** 2

**Q1 Summary And Contributions:**

This paper is an application paper which uses deep learning in order to improve Adaptive Optics (AO) systems. AO relies on the control of deformable mirors. However, there is a residual turbulence in the images which leads to optical abberations. The authors propose a (real-time) GAN method which estimates the residual perturbation of WFS (wavefront sensor) images by splitting the WFS image between a linear and non-linear part. The authors evaluate their GAN model on simulated data.

**Q2 Assessment Of The Paper:**

More detailed information regarding each of these aspects is given below:

**Q2(4) Quality Of Experiments (Optional):**

2: Fair: The experimental evaluation is weak: important baselines are missing, or the results do not adequately support the main claims.

**Q2(5) Reproducibility:**

1: Poor: Key details (e.g., proof sketches, experimental setup) are incomplete/unclear, or key resources (e.g., proofs, code, data) are unavailable.

**Q3 Main Strengths:**

-I really enjoyed the experimental setting of this paper. Applying deep learning techniques to astronomical observations is an exciting challenge. I think that (for the most part, see the detailed comments), the authors did a good job at introducing the basics of Adaptive Optics.

-The training of GAN for such data and the application in real-time to real-world problem is promising. The simplicity of the method also shows that this method could be adopted in practice.

**Q4 Main Weakness:**

-I think that, even though the paper does a great job at introducing astronomical quantities, the quality of the writing is not as good when it comes to the description of the algorithm. I found it hard to understand what was the real input of the GAN. Reading Section 5.3 I understood that "taking the closed loop WFS image using the GAN to infer the residual phase" but it's not clear reading Section 4. I think that the paper would benefit from a pseudo-code algorithm (and releasing the code).

-I find a little bit concerning the absence of strong baselines (expect the linear closed-loop) to assess the efficiency of the method. In particular, the second DM could be controlled with a method which is not based on deep learning algorithm [1]. However, if I'm not mistaken, the method is only evaluated against the "Linear closed-loop" method which only assumes access to one DM.

[1] Gavel, Norton --Woofer-tweeter deformable mirror control for closed-loop adaptive optics: theory and practice

**Q5 Detailed Comments To The Authors:**

-At the moment I think the paper is still to hard to read for the major part of the ML community. While the description of the model is good and useful, key details (pseudo-code or at least clarifications of the inputs, reported ablation studies) are missing in the paper.

-The notions of closed loops and open loops are used through the paper but are never recalled. I think it would be very useful to recall these notions so that the difference between the two loops is clear.

-Figure 1 (caption):  "purturbations" --> "perturbations"

-"meaurement wavelength, etc.) Model-based approaches" (missing period)

-In Equation (4) --> log --> \log

-Figure 5, could it possible to show the range of these images? (Is it the same on the left hand side and on the right hand side?)

-I think the discussion could be shortened to 1) give more details on the machine learning model used by the authors 2) give more details on Equations (2) and (3) (especially how the matrices R and R^nl are derived).

-The code is not available currently and no indication is given regarding a future release.

**Q7 Justification For Your Score:**

Even though AO is not my area of expertise, I enjoyed reading the paper which I think could be of interest to the broad community of UAI. However, I think the paper needs substantial rewriting (regarding the algorithmic details) in order to fit to the UAI standards. I also wish the authors had evaluated their method against stronger baselines.

**Q9 Complying With Reviewing Instructions:**

1: Yes.

---

### Official Review · Reviewer_p7w1 · 2022-04-12

**Q2(1) Originality/Novelty:** 2
**Q2(2) Significance/Impact:** 4
**Q2(3) Correctness/Technical Quality:** 3
**Q2(6) Clarity Of Writing:** 3
**Q6 Overall Score:** 7
**Q8 Confidence In Your Score:** 3

**Q1 Summary And Contributions:**

The paper exploits a significant opportunity in adaptive optics, namely the estimation of higher-order wavefront information and its use in deformable mirror control through actuators, which, through a purely software-based solution, without any hardware or infrastructure upgrade, increases the resolution of images which can be acquired on Earth-based telescopes. The solution offered can take advantage of more actuators where available.

**Q10 Ethical Concerns (Optional):**

No.

**Q2 Assessment Of The Paper:**

More detailed information regarding each of these aspects is given below:

**Q2(4) Quality Of Experiments (Optional):**

3: Good: The experimental evaluation is adequate, and the results convincingly support the main claims.

**Q2(5) Reproducibility:**

2: Fair: Key resources (e.g., proofs, code, data) are unavailable but key details (e.g., proof sketches, experimental setup) are sufficiently well-described for an expert to confidently reproduce the main results.

**Q3 Main Strengths:**

The experimental validation is convincing and demonstrates very good results, both against baselines (linear closed loop) and in different operating conditions.
The neural architecture adopted here (GAN with UNet and PatchGAN as discriminator, where the conditioning input is the WFS image and the generated output is the wavefront phase, which directly guides the actuators) is adapted. It is shown to be robust against change in turbulence regimes (r_0 ranges table 2), which brings an extra advantage.
The design of the simulation using COMPASS seems to work well and seems to correspond to standards in AO, as it is easy to integrate, and simulates air turbulence and its optical effects with high quality.
The paper seems to hesitate between exposing to an AO audience or to an ML audience, prudently explaining both aspects with quite some background. This has the advantage of making the paper's ideas accessible, but also reduces space available for more specialist discussions.


**Q4 Main Weakness:**

My main uncertainty is about the relevance of this paper to the UAI target audience. As far as I can assess, it does represents a significant improvement in adaptive optics, and therefore is a strong example of the impact of applying uncertainty estimation methods in experimental physics. It does not bring major innovations in machine learning as the models used are, by the words of the authors, established models, to which quite few modifications are brought.
Ablation studies are mentioned sec4 Data Transformations, but are absent from the paper and supplementary material.


**Q5 Detailed Comments To The Authors:**

For the UAI audience, more emphasis could be given to ML aspects, especially considering that space can be saved by cutting background on GANs in general, the UNet design, conditional GANs, which must be assumed to be familiar to the UAI readership.
It is hard to work out exactly which data are processed by the GAN's different modules (as opposed to the control loops sec3); in ML papers, usually data/quantities are introduced formally including their dimensionality, structure, range. The only informative equation is eq4-6, which is really textbook material (with the exception maybe of the idea of using a loss on a masked output).

**Q7 Justification For Your Score:**

I weighed more heavily the impact and relevance as a convincing applied solution. I gave less importance to the only incremental degree of innovation in ML terms.

**Q9 Complying With Reviewing Instructions:**

1: Yes.

---

### Official Review · Reviewer_cMsR · 2022-04-16

**Q2(1) Originality/Novelty:** 2
**Q2(2) Significance/Impact:** 2
**Q2(3) Correctness/Technical Quality:** 3
**Q2(6) Clarity Of Writing:** 4
**Q6 Overall Score:** 5
**Q8 Confidence In Your Score:** 4

**Q1 Summary And Contributions:**

This paper proposes to use a GAN-based image-to-image translation model to enhance the adaptive optics (AO) control in telescopes. The GAN-based model is used for phase estimation for AO control. The experiments on synthetic data show that the GAN based method can estimate more accurately than conventional methods.

**Q2 Assessment Of The Paper:**

More detailed information regarding each of these aspects is given below:

**Q2(4) Quality Of Experiments (Optional):**

2: Fair: The experimental evaluation is weak: important baselines are missing, or the results do not adequately support the main claims.

**Q2(5) Reproducibility:**

3: Good: Key resources (e.g., proofs, code, data) are available and key details (e.g., proofs, experimental setup) are sufficiently well-described for competent researchers to confidently reproduce the main results.

**Q3 Main Strengths:**

- This paper is well written and organized. It is very easy to follow.
- The work is well motivated.
- The GAN based method is properly integrated into the AO control system and the corresponding phase estimation problem.
- The authors present the background knowledge and the whole big picture (of the system) in detail.


**Q4 Main Weakness:**

- The main contribution is an application of an existing method to a scientific problem. The technical contribution on model design is limited.
- In experiments, this paper only studies one particular GAN based model. The influence of the choice on the model structures is not considered and studied. Even though the model design is not the core contribution, studies on how the specific model influences the performance are very valuable for the scientific problem.
- The experiments are restricted to synthetic data.
- Apart from only the phase estimation accuracy, the authors may study how the estimation error will influence the control task.


**Q5 Detailed Comments To The Authors:**

Please check weakness.

**Q7 Justification For Your Score:**

It is a significant contribution to apply the existing ML/DL techniques to the scientific problem.

The main contribution is the application of existing AI techniques in a scientific task.
The contributions on AI techniques are limited.

The MAIN drawback is that how the choice of the technique details influences the task is not studied. It is essential for such kind of work, even though there is no new models. And experiments are restricted to synthetic data.


**Q9 Complying With Reviewing Instructions:**

1: Yes.

---

### Decision · Program_Chairs · 2022-05-15

**Decision:**

Accept (Poster)

**Comment:**

Meta Review: The authors describe a GAN model for estimating perturbations in wavefront sensor images for optical telescope acquisition of astronomy data.  They incorporate their model into an adaptive optical feedback mechanism and show significant improvements over the baseline.

The contribution of the paper is mainly application-oriented, showing a proof of concept and promise of the authors' approach.  The models themselves are not novel, and the authors do not explore a variety of alternatives (although they do discuss this in more detail in the author response).  That said, however, the paper is a solid application, with non-trivial domain knowledge required to develop a solution that can be integrated into the adaptive control mechanism, and the empirical assessment is thorough and convincing.  Author responses addressed many of the initial reviews' concerns.  Overall this is a nice application paper.